# Comprehensive Thermodynamic Analysis of the Humphrey Cycle for Gas Turbines with Pressure Gain Combustion

**Panagiotis Stathopoulos**

Chair of Unsteady Thermodynamics in Gas Turbine Processes, Hermann Föttinger Institute,
Technische Universität Berlin, Müller Breslau Str. 8, 10623 Berlin, Germany; stathopoulos@tu-berlin.de;
Tel.: +49-30-314-29361

**Abstract:** Conventional gas turbines are approaching their efficiency limits and performance gains are becoming increasingly difficult to achieve. Pressure Gain Combustion (PGC) has emerged as a very promising technology in this respect, due to the higher thermal efficiency of the respective ideal gas turbine thermodynamic cycles. Up to date, only very simplified models of open cycle gas turbines with pressure gain combustion have been considered. However, the integration of a fundamentally different combustion technology will be inherently connected with additional losses. Entropy generation in the combustion process, combustor inlet pressure loss (a central issue for pressure gain combustors), and the impact of PGC on the secondary air system (especially blade cooling) are all very important parameters that have been neglected. The current work uses the Humphrey cycle in an attempt to address all these issues in order to provide gas turbine component designers with benchmark efficiency values for individual components of gas turbines with PGC. The analysis concludes with some recommendations for the best strategy to integrate turbine expanders with PGC combustors. This is done from a purely thermodynamic point of view, again with the goal to deliver design benchmark values for a more realistic interpretation of the cycle.

**Keywords:** pressure gain combustion; gas turbine; Humphrey cycle; thermodynamic analysis; turbine cooling; turbine integration

## 1. Introduction

Based on information from the International Air Transportation Association [1], 3.8 billion passengers traveled by air in 2016, which is 8% more than the previous year. The Organization for Economic Cooperation and Development forecasts that air transport $CO_2$ emissions will grow by 23 % by 2050, if no measures for their abatement are taken [2]. Considering this, stringent environmental regulations are already in place with the ultimate goal to cut net emissions to half of the 2005 level by 2050. It is for this reason that engine manufacturers focus on possible ways to increase engine efficiency. At the same time, stationary gas turbines are the only thermal power plant technology capable of delivering both secondary and tertiary control reserve from idle [3]. The rapid expansion of renewable generation in Europe is expected to double the demand for both reserves in the coming decade [4]. If one considers that gas turbines are very likely to be able to convert hydrogen into electricity at a large scale, an increase in their efficiency can prove very valuable on the road towards carbon free power generation.

Pressure Gain Combustion (PGC) has the potential to increase the propulsion efficiency of aero-engines and the thermal efficiency of stationary gas turbines. Up to date, detonative combustion processes have been the primary method to realize pressure gain combustion, such as pulsed [5] and rotating detonation combustion [6], with the latter gaining more attention. Two alternative approaches

are the shockless explosion combustion [7] and pulsed resonant combustion [8]. Both use resonant pressure waves in a combustor to realize quasi constant volume combustion. The ideal thermodynamic cycles that model gas turbines with pressure gain combustion are the Humphrey and the ZND cycle, presented in Figure 1 along with the Joule cycle. The Humphrey cycle models gas turbines with ideal constant volume combustion and is best suited for the cases of shockless explosion combustion and resonant pulsed combustion. The ZND cycle models the application of detonative combustion in gas turbines.

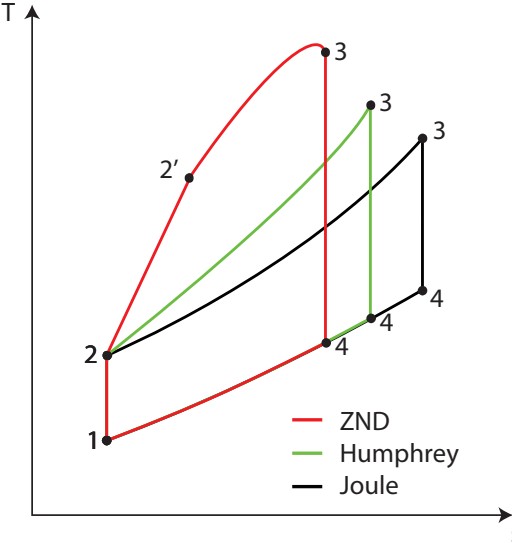

**Figure 1.** T-s diagrams of the Joule, Humphrey and ZND cycles.

Heiser and Pratt [9] were the first to theoretically demonstrate the potential of pressure gain combustion to raise the efficiency of gas turbines. Their analysis focused on the ideal Humphrey and ZND cycles and concluded that the main reason for their higher efficiency is the lower entropy increase during combustion. By extending their analysis to include the turbomachinery isentropic efficiency, they showed the importance of the expander efficiency for the cycle efficiency. However, the T-s diagrams and the respective ideal cycle calculations do not model the actual physical phenomena in pressure gain combustion systems in a satisfactory way. In fact, the processes are periodic and time-dependent in the combustor, while they can be easily represented by quasi steady values in time at the outlet of the compressor. Another very important aspect of the analysis in [9] was the assumption that expansion, and thus work extraction from the working medium, starts at the highest temperature point of the cycle (point 3 in Figure 1). Nalim [10] indicated both shortcomings. He thus proposed a simplified model that accounted for the internal expansion in a pressure gain combustor. This model delivers an equivalent steady thermodynamic state at the outlet of constant volume combustors that can be then used for an analysis similar to that in [9]. This model does not account for entropy generation due to shock in detonations, but it is a good approximation of the physical phenomena taking place in the two PGC technologies, which are best modeled by the Humphrey cycle (see Section 2.1). Paxson et al. [11] proposed a more detailed way to account for the time variation at the outlet of pulsed detonation combustors. They used a typical operational map of a turbine expander and computed the work output with a quasi steady-state model. This approach has been adopted in the work of Stathopoulos [12] and Rähse [13,14] to compute the thermal efficiency of the pulsed detonation and the shockless explosion cycles. In this case, the processes in the combustor are resolved in time by solving the 1-D time dependent Euler equations with source terms for the chemical reaction. The time-resolved combustor outlet conditions were then fed to a turbine expander model that computed the generated work in a similar way as in [11]. Nordeen applied a similar method to resolve the outlet conditions in a rotating detonation engine, also with the aim to compute the thermodynamic

efficiency of the cycle [15]. Irrespective of the type and approach of the aforementioned models, effects such as detonation-to-deflagration transition, quasi constant volume combustion and the pressure drop at the combustor inlet and outlet have not been accounted for in a holistic manner.

The exhaust flow of pressure gain combustors is characterized by strong pressure, temperature and velocity fluctuations [16,17]. The main challenge in the practical implementation of PGC into gas turbines is the lack of turbomachinery that can efficiently harvest work from the PGC exhaust gas. Although still a topic of active research, it is generally accepted that conventional turbine expanders have a lower isentropic efficiency when they interact directly with pressure gain combustors [18,19]. To address this challenge, one can follow two extreme methods. According to the first, a plenum or combustor outlet geometry could be designed to adapt the exhaust stream of a PGC to an extent that it could be fed to a conventional turbine. In this case, the latter would operate at its design efficiency. The other approach focuses on a dedicated turbine design that could directly expand the outlet flow of a PGC. A much more rational approach would be to optimize the combination of a PGC outlet geometry and an adapted turbine design to achieve the maximum possible work extraction. The current work aims at benchmarking the latter approach for the cases of shockless explosion combustion and pulsed resonant combustion. In this way, insights on the allowable limits for the losses in exhaust gas conditioning devices and the maximum allowable reduction in turbine efficiency can be gained.

Another aspect of the cycles that has been neglected in all previous thermodynamic evaluations is turbine cooling. This topic has two implications. On the one hand, the combustor is expected to deliver an average pressure increase over a limit cycle. This implies that the cooling air for the first turbine stage has to be compressed by an additional compressor. On the other hand, turbine cooling reduces the cycle efficiency for the same turbine inlet temperature and its effect on PGC gas turbine cycles has not been analyzed yet. Furthermore, it has been shown by numerous studies on turbine integration that the pressure, velocity and temperature fluctuation stemming from PGC combustors are largely attenuated through the first turbine stage [18,20]. This means that the remaining turbine stages will most probably work at their nominal isentropic efficiency. Up to date, the expansion efficiency has been lumped in one equivalent efficiency of the whole turbine. The current work aims at resolving this issue and its impact on cycle efficiency.

In summary, the present work aims at resolving several open questions on the Humphrey gas turbine cycle. More specifically, the effect of excursions from ideal constant volume combustion on the cycle and its thermal efficiency are explored. The current work is also the first that accounts only for reductions in the the first turbine stage efficiency and thus clarifies the demand for further research in the field of turbine design. In the same scope, the sensitivity of the cycle efficiency on the installation of exhaust gas conditioning devices at the turbine inlet is studied. The current work also aims at clarifying the importance of turbine cooling for the efficiency of the Humphrey cycle, as it is compared to an equivalent Joule cycle with turbine cooling. Moreover, the impact of an additional compressor that delivers cooling air to the first turbine stage is analyzed.

To answer these questions, a new steady state model of the Humphrey cycle was developed in Aspen plus, the details of which are presented in Section 2. Section 3 presents the results of the analysis, and the current work concludes with some recommendations for further work on the attempted cycle analysis.

## 2. Methods and Modeling Approach

### 2.1. Combustor Model

There are several ways to model pressure gain combustion for the thermodynamic analysis of the respective cycles. Depending on the aim of the analysis, one could attempt to resolve all thermodynamic and gas dynamic processes in a combustor and thus resolve the sources of all losses in detail. This has been the approach of several studies dedicated on understanding detonative

combustion, be it pulsed detonation [16] or rotating detonation [21–23]. In the current work, the model of Nalim [10] has been chosen to represent the pressure gain combustion process of the Humphrey cycle. Based on this model, PGC is modeled as a constant volume combustion process, the products of which expand eventually to atmospheric pressure. Part of the expansion process takes place inside the combustor with no work generation, while another part happens in a turbine expander and generates work. The model can in this way deliver an equivalent steady thermodynamic state at the outlet of a periodic pressure gain combustion chamber, as in pulsed resonant combustors or shockless explosion combustors. This thermodynamic state is subsequently used to model the Humphrey cycle as an open, steady heat engine cycle.

Figure 2 presents the basic thermodynamic states of the combustor model. In the current representation, we do not take the existence of any buffer gas into account and we neglect any possible pre-compression of the combustible mixture. This leaves us with three thermodynamic states. The first (A-2) represents the combustor inlet and is the same as states A and 2 of the model presented in [10]. State B is the working medium state at the end of a constant volume heat addition process. It is assumed that the working medium commences its expansion process from that state to atmospheric pressure. Part of this expansion takes place in the combustor itself (B-3), and is necessary to expel the products from it, and part in a turbine expander (3-4). The model thus delivers the outlet temperature and pressure of the combustor (state 3) based on an isentropic expansion process B-3. In the current work, the conditions at point three are considered the inlet conditions of the three stage turbine, the model of which is presented in Section 2.3.

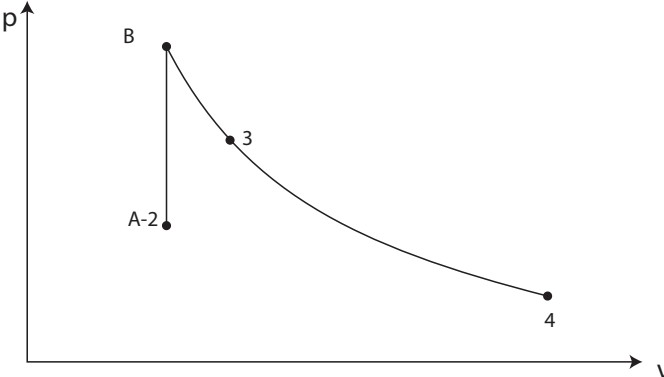

**Figure 2.** Pressure-specific volume diagram of the combustor model.

By assuming constant material properties (chosen at the average temperature and pressure expected in the combustor chamber), one can first compute the pressure and temperature at point B. For that, pure constant volume heat addition is assumed and the first leg of Equation (1) is applied. By applying the energy conservation over the whole combustor, its outlet temperature $T_3$ can be computed through the second leg of Equation (1).

$$Q = \dot{m} \cdot c_v \cdot (T_B - T_A) = \dot{m} \cdot c_p \cdot (T_3 - T_A) \tag{1}$$

In Equation (1), Q is the total heat input in the combustor, based on the lower heating value of the fuel, whereas $\dot{m}$ is the total mass flow entering it. In the current work, only hydrogen is considered as fuel. From the simple assumption of a perfect gas in the combustion chamber, the pressure change during constant volume heat addition can be computed by Equation (2).

$$\frac{T_B}{T_A} = \frac{p_B}{p_A} \tag{2}$$

From the pressure and temperature values at point B and the temperature ratio $\frac{T_B}{T_3}$, one can subsequently approach the combustor internal expansion with an isentropic process and compute its mass averaged outlet pressure with Equation (3).

$$\frac{p_3}{p_B} = \left(\frac{T_3}{T_B}\right)^{\frac{\gamma}{\gamma-1}} \tag{3}$$

Equations (2) and (3) can be used to compute an equivalent thermodynamic state at the outlet of a pressure gain combustor, based on its inlet conditions and its fuel. However, pressure gain combustors typically demonstrate a slightly different behavior from that expected in an ideal case. In pulsed detonation combustion, this corresponds to the fuel consumed during the deflagration-to-detonation phase. In rotating detonation combustors, part of the mixture is consumed through contact burning directly after the recovery phase of the mixture injection [15]. In a pulsed resonant combustor, the partial confinement of the combustible mixture is the main source of discrepancies. Finally, perturbations cause small but important departures from the ideally defined shockless explosion combustion process [24].

To model these effects, part of the combustible mixture is assumed to be combusted under constant pressure conditions. Hence, the described PGC combustor is divided in two components (see also Figure 3), one functioning as a pure PGC combustor and a second one that is a constant pressure combustor. When introducing partial constant pressure combustion, it is necessary to chose the pressure at which this combustion process takes place. In most actual processes, constant pressure precedes pressure gain combustion. For example, in pulsed detonation combustors, quasi-constant pressure combustion takes place at the inlet pressure of the combustor until the deflagration-to-detonation process ends and pressure gain combustion commences. The mass of the mixture combusted during this phase goes through subsequent compression and expansion process, which are closely connected with the gas dynamic phenomena in the combustion chamber. The complexity of these phenomena does not allow their easy integration in a simplified combustor mode, such as the one proposed here. For this reason, it is assumed that the constant pressure combustor of Figure 2 operates at the peak pressure of the PGC combustor (i.e., $p_B$ in Figure 2). Its products are then also isentropically expanded to the outlet pressure of the PGC combustor (i.e., $p_3$ in Figure 2). In this way, a realistic but simplified representation of the actual complex physical phenomena is achieved. The two outlet streams are mixed without any losses and the resulting temperature is assumed to be the turbine inlet temperature for the rest of the cycle. The percentage of the mixture that burns in the constant pressure combustor is a free variable for the general cycle model. Its influence on cycle performance is analyzed in detail in Section 3.

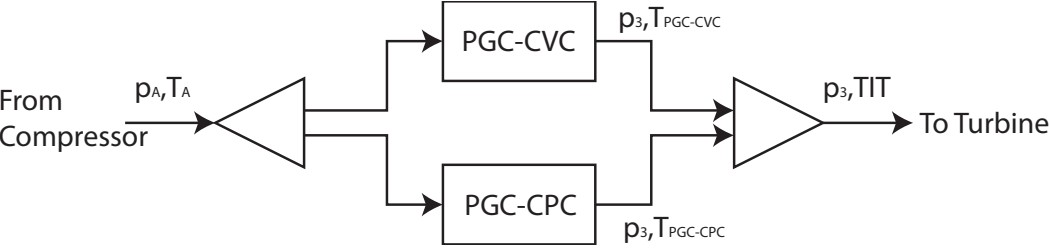

**Figure 3.** Schematic of the two component of the PGC combustor and their connections to the upstream and downstream components of the cycle.

## 2.2. Secondary Air System Model

The applied secondary air system model is based on the work of Kuzke [25] and Horlock [26]. It computes the cooling air mass flow rates for each turbine blade row and the pressure losses associated to mixing processes. In the following, it is described for one of the three turbine stages.

Figure 4 presents the h-s diagram of the expansion in a cooled turbine stage. The process has three steps. In the first step, the turbine stator cooling air is mixed to the main exhaust gas stream before its expansion in the rotor. The pressure drop due to the mixing process is taken into account by an appropriate pressure loss coefficient (see section 2.3). The expansion of the resulting stream in the rotor is modeled in the same way as in an uncooled expander. Finally, the rotor cooling air stream is mixed with the expanded gas at the outlet of the rotor blade and generates work only in the succeeding turbine stage.

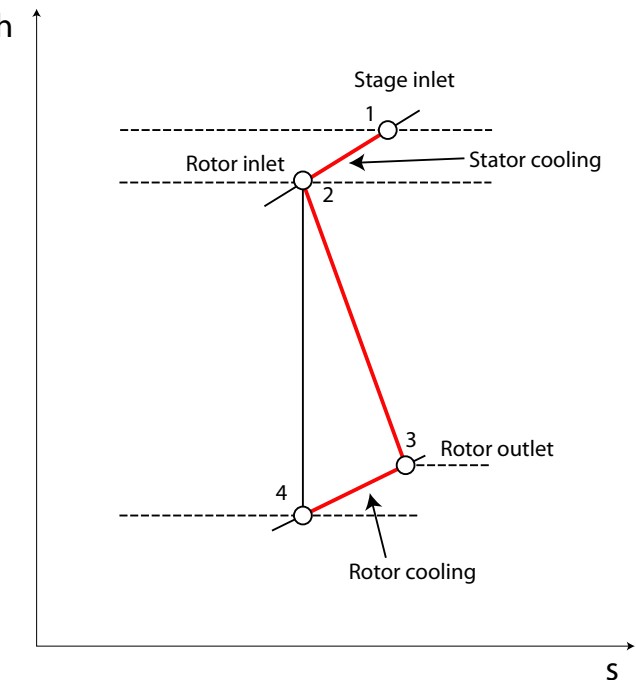

**Figure 4.** Specific enthalpy–specific entropy diagram of the applied expansion model in a cooled turbine stage after [25,26].

The cooling air flow rates are computed based on the assumption that the external Stanton number $St_g$, remains constant as the gas turbine design parameters and the turbine inlet pressure and temperature change [26]. The chosen cooling system technology level, its effectiveness and a pressure loss coefficient are presented in Table 1. The blade material temperature is kept constant and equal to 1100 K throughout the computations of the Humphrey cycle and the equivalent Joule cycle.

**Table 1.** Cooling system model parameters based on the model from [26].

| Parameter | Symbol | Value |
|---|---|---|
| Cooling efficiency | $\eta_{Cooling}$ | 0.9 |
| Film cooling effectiveness | $\varepsilon_F$ | 0.4 |
| Level of technology constant | C | 0.045 |
| Pressure loss constant | K | 0.07 |

In addition to these calculations, the expansion ratio of each turbine stage ($\delta p_{stage}$) must be defined. In the current work, the total expansion ratio from the outlet of the PGC combustor to atmospheric pressure is equally distributed among the turbine stages. Based on this assumption, only the first three blade rows have to be cooled. Another particularity of the turbine cooling air system is that the pressure at the inlet of the first stage is higher than that at the outlet of the compressor. This is overcome by the installation of an additional smaller compressor that delivers cooling air at the first stator row of the turbine. It must be stressed here that the cooling air of the rotor row is mixed to the

main exhaust stream after its expansion. Hence, there is no need for an additional compressor for this cooling air stream.

### 2.3. Gas Turbine Model

This section deals with the cycle component models that have not been described in Sections 2.1 and 2.2. The following basic assumptions formed the general framework of the presented simulations:

- The working fluid was considered a real gas and its properties were computed by the Aspen Properties database (the RK-BS model has been used for that). Only the processes in the combustion chamber were computed with the average properties at its inlet and outlet.
- The compression process was adiabatic with a given constant isentropic efficiency.
- The combustion products (and not air) were taken as the working fluid of the turbine expander.

Aspen plus was used for the simulation of the gas turbine operation. The main reason was its comprehensive database for material properties and our extensive experience at its implementation in gas turbine models and applications [27,28]. However, the software does not provide a model for the PGC combustor used in this work. To solve this, a user defined function was developed for the combustor and then integrated into the cycle model. The secondary air system equations (see Section 2.2) were integrated in the Aspen plus model with the help of calculator modules.

Figure 5 presents the model schematic with the most important components. The combustor is a "black box" containing all the components presented in Figure 2. The mixing component before the combustor was used to model pressure losses at its inlet. Only the first three turbine blade rows were cooled, whereas only the cooling air for the first stator row must be compressed to the combustor outlet pressure. This was a result of the assumption that the cooling air for the rotor is mixed to the main exhaust flow after the expansion process. Thus, the exhaust was already at a pressure below the outlet pressure of the compressor and there was no need for any additional compression of the cooling air. This assumption slightly underestimated the work consumption of the additional compressor, since part of the cooling air was injected as film cooling in the leading edge and along the surface of the blades. Each cooled turbine stage was modeled with two components: a mixing element for the stator and an expander element for the rotor. Based on the model presented in Figure 4, only the pressure drop due to the cooling air mixing with the main exhaust stream at the outlet of the stator was accounted for. The total pressure loses in the stator were lumped in the isentropic efficiency of the expander element that followed. Moreover, each expander element had its own efficiency. The schematic diagram of the applied turbine model in Apsen plus is presented in Figure 6. The last uncooled turbine stage was represented by a simple expander element with a fixed isentropic efficiency.

Table 2 presents the model parameters in detail along with the chosen expansion ratio distribution for the turbine stages.

**Table 2.** Model parameters and assumptions.

| Component | Symbol | Value |
|---|---|---|
| Compressor | $\eta_{isC}$ | 0.9 |
| Cooling air compressor | $\eta_{isC-cool}$ | 0.9 |
| Turbine | $\eta_{isT1}$ | 0.7–0.9, variable |
| | $\eta_{isT2}$ | 0.9, fixed |
| | $\eta_{isT3}$ | 0.9, fixed |
| | $\delta p_{stage}$ | $p_3^{-\frac{1}{3}}$ |
| Cooling system | $T_{bl}$ | 1100 K |
| | $\delta p_{mix}$ | 1% |

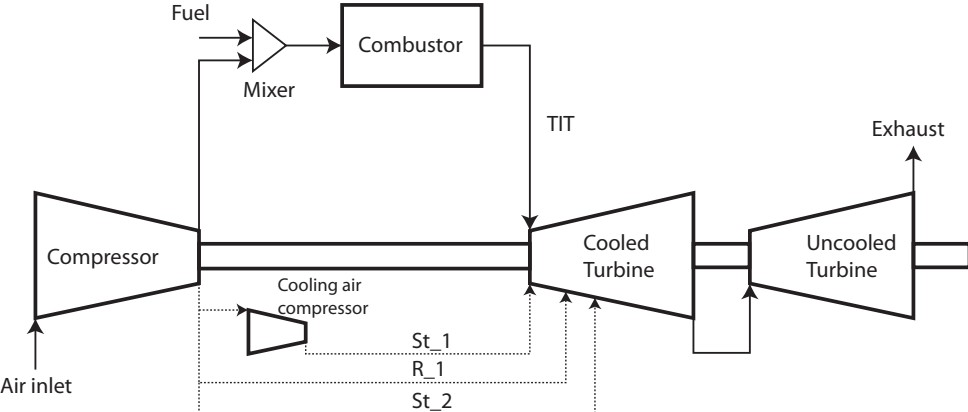

**Figure 5.** Schematic of the cycle model with the most important components, St_1, R_1 and St_2 refer to the first stator and rotor rows and the second stator row respectively.

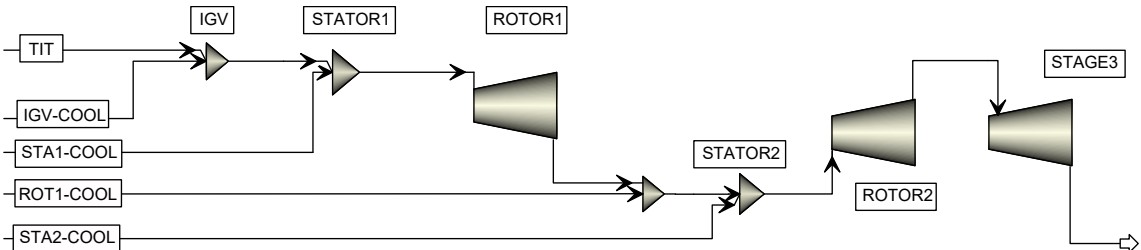

**Figure 6.** Aspen plus flow sheet of the applied turbine model and the case with IGVs (see Section 3.1.5).

### 2.4. Simulation Procedure

Gas turbine thermodynamic cycles are usually compared either for the same dimensionless heat addition [29] or at the same turbine inlet temperature. The former approach is applied to the majority of PGC gas turbine thermodynamic cycles, even though it results in different turbine inlet temperatures (TIT). In the current work, two values of the turbine inlet temperature werechosen to compare the Humphrey cycle to the Joule cycle. In this way, realistic values for the cooling air flows and the respective equivalence ratios for each combustion system could be chosen. Once the TIT values were chosen, the compressor pressure ratio was varied between 10 and 40 and the thermal efficiency was computed along with other characteristic variables of each cycle. The pressure ratio region was chosen based on the most typical values found in stationary gas turbines. Pressure ratio values below 10 can typically be found in smaller and older systems, which are outside the scope of the current work.

The first set of simulations was performed with the assumption that 100% of the mixture entering the combustion chamber participated in pressure gain combustion. The pressure drop at the inlet of the combustor was chosen equal to 5% of the inlet pressure, whereas all turbine stages had an isentropic efficiency of 90%. In the second set of computations, 20% of the mixture mass was combusted under constant pressure conditions, while all other cycle parameters where kept the same. In this way, first insights could be gained on the effect of the deflagrative combustion part on the cycle parameters. In the third case of simulations, the pressure drop at the combustor inlet (that across the mixing element in Figure 5) was increased to 15% of the inlet pressure, while everything else was kept the same as in Case 2. This simulation aimed at providing a first impression of the effect of inlet pressure drop on cycle efficiency. The first series of simulations was concluded by reducing the isentropic efficiency of only the first turbine stage. This was done because the vast majority of turbine integration studies with PGC have shown that the pressure, velocity and temperature fluctuations at the combustor outlet are strongly attenuated after the first turbine stage [20,30,31]. It is thus fair to assume that the remaining two turbine stages would operate at their nominal point and no further turbine efficiency reduction should be considered. Table 3 presents the settings for the studied parameters in the first series of simulations.

The values of the three parameters in question for Case 4 in Table 3 are considered fairly realistic. Based on this setup, a reference gas turbine design has been defined, from which the subsequent sensitivity studies started in the second series of simulations. The latter have the goal to thoroughly study the influence of each parameter on cycle performance. In each sensitivity study all parameters apart one where kept constant to the values presented in Table 4. One parameter was then varied in a range of values that allowed to observe its effect on the chosen cycle parameters. Table 4 presents the extend, to which the variables in question have been changed during the sensitivity analysis.

**Table 3.** Values of the parameters for the first stage of simulations.

| Case | Mass through CPC (%) | Inlet Pressure Drop, % of $p_{in}$ | $\eta_{isT}$, % |
|------|----------------------|-----------------------------------|------------------|
| Case 1 | 0 | 5 | 0.9 |
| Case 2 | 20 | 5 | 0.9 |
| Case 3 | 20 | 15 | 0.9 |
| Case 4 | 20 | 15 | 0.7 |

**Table 4.** Values of the parameters for the sensitivity analysis.

| Parameter | Symbol | Reference Value | Variance Region |
|-----------|--------|-----------------|-----------------|
| Mass through CPC (%) | $m_{CPC}$ | 20 | 5–40 |
| Combustor inlet pressure drop, % of $p_{in}$ | $dp_{CC}$ | 15 | 0–30 |
| Turbine first stage isentropic efficiency, - | $\eta_{isT}$ | 0.75 | 0.6–0.9 |
| IGV pressure drop, % of $p_{in}$ | $dp_{IGV}$ | - | 5–20 |

As already mentioned, there are two ways to approach energy/availability harvesting from the exhaust of a pressure gain combustor. One approach is to develop a turbine design that directly expands the outlet flow of a PGC. According to literature, this choice will most probably have a strong impact on the isentropic efficiency of the first turbine stage. This approach is covered by the isentropic efficiency sensitivity analysis, presented in Table 4. Another approach would be to install a plenum or combustor outlet geometry and condition the exhaust gas so that it could be fed to a conventional turbine. In this case, the latter would operate at its design efficiency. This case is covered in the current work by carrying out an additional set of simulations, where such a combustor outlet geometry is modeled as a cooled turbine inlet guide vanes (IGV) row. In this case, the IGV row results in a given pressure drop and the turbine stages downstream operate at their nominal isentropic efficiency (i.e., 0.9). The effect of this additional blade row is studied through a sensitivity analysis that has Case 3 (see Table 3) as its starting point and adds a row of blades with varying pressure drop, the values of which are shown in Table 4.

## 3. Cycle Analysis Results

### 3.1. Basic Thermodynamic Analysis

This section presents the results of the cases presented in Table 3. The aim is to start from a rather optimistic cycle setup in Case 1, where the pressure drop at the combustor inlet is comparable to the total pressure drop in a conventional gas turbine combustor (i.e., 5%). At the same time, Case 1 considers no mass consumption in the deflagrative part of the combustor. Finally, Case 1 looks at the most optimistic representation of turbine efficiency, since the turbine has a constant isentropic efficiency of 0.9 and no conditioning device is considered between the turbine and the combustor exit. All in all, Case 1 is considered the upper efficiency limit for the studied cycle. A part of this ideal representation of the cycle is taken away by assuming that 20% of the mass that enters the combustor is burned under constant pressure in the scope of Case 2. Case 3 makes the next step through the introduction of a higher pressure drop at the combustor inlet. Although experimental pressure gain combustors have been operated with considerable higher pressure drops at their inlet, their operation

with this pressure drop value could be feasible. Finally, Case 4 looks at the effect of turbine efficiency deterioration, due to the time variation of its inlet conditions. This is done by reducing its isentropic efficiency from 0.7 to 0.9.

### 3.1.1. Thermal Efficiency Results

Figures 7 and 8 present the thermal efficiency as a function of the compressor pressure ratio for two representative turbine inlet temperatures and the four cases from Table 3, along with the results for the Joule cycle.

In Figure 7, it is obvious that for a TIT of 1300 °C even the most optimistic case of the Humphrey cycle (PGC1—Case 1 in Table 3) would result in a modest increase of efficiency up to a pressure ratio of 32. This efficiency gain is rather small for pressure ratios between 20 and 32 and becomes considerable for lower pressure ratios. It reaches its maximum value at the lowest investigated pressure ratio, where an increase of 5% is observed. As expected, the cycle efficiency is reduced when deflagrative combustion and the combustor inlet pressure drop are considered, thus making the use of the cycle at this TIT questionable. Finally, no efficiency increase against the equivalent Joule cycle is expected if the turbine isentropic efficiency drops from 0.9 to 0.7 (see Case 4).

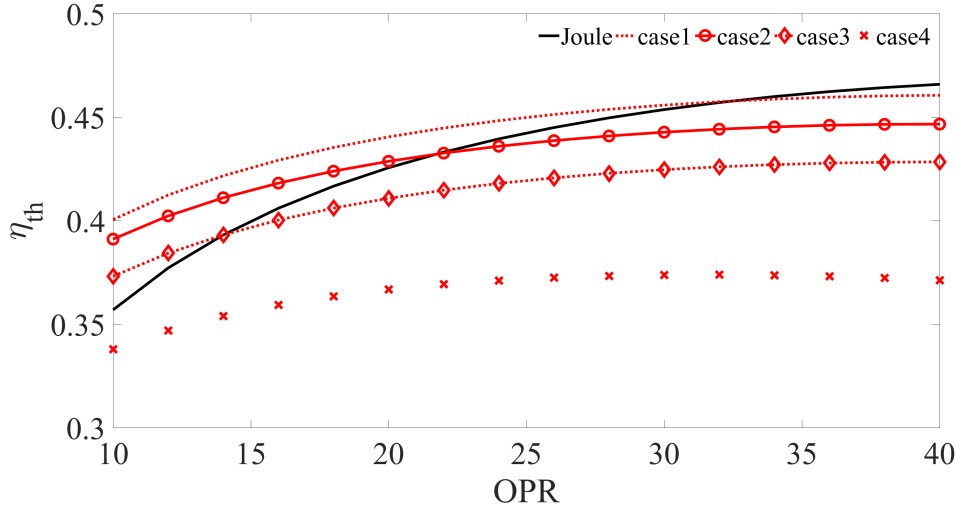

**Figure 7.** Cycle thermal efficiency as a function of compressor pressure ratio for the Joule cycle and the four versions of the Humphrey cycle presented in Table 3. TIT: 1300 °C.

This picture is considerably changed through an increase in the turbine inlet temperature to 1700 °C, as shown in Figure 8. In this case, it becomes clear that an efficiency advantage against the Joule cycle is possible for the Humphrey cycle if the turbine isentropic efficiency is higher than 0.7. In fact, Cases 1–3 show an efficiency increase for almost all investigated cycle pressure ratios. Again, the Humphrey cycle demonstrates a larger efficiency advantage for relatively low pressure ratios, while its advantage diminishes for higher values. Even Case 4 shows some small efficiency advantage for the lowest pressure ratios and reaches efficiency parity with the Joule cycle at a pressure ratio of approximately 12.

There are several ways to explain the presented efficiency results. On the one hand, the outcome is expected to some extent, based on previous studies of similar but simpler cycle configurations [9,11,12]. The reduction of the efficiency advantage for higher pressure ratios is also expected due to the higher expansion ratio in the turbine. This is already known to make PGC cycles generally more sensitive to changes in the turbine efficiency [9]. However, several other cycle parameters have an influence in the current case.

Figure 9 presents the values of the combustor pressure ratio and its equivalence ratio. As expected from Equation (2), there is a direct connection between the equivalence ratio and the pressure gain

across the combustor. This is a known effect, since the larger is the specific heat addition to a constant volume or pressure gain combustor, the higher is the pressure increase. At the same time, the equivalence ratio is connected to the final outlet temperature of the combustor through the energy balance across it. This connection explains the decrease of the global equivalence ratio for increasing cycle pressure ratios. Higher compressor pressure ratios lead to higher compressor outlet temperatures. For a constant TIT, a higher combustor inlet temperature will result in lower equivalence ratios, and hence lower combustor pressure ratios. The decreasing combustor pressure gain for increasing cycle pressure ratios is another reason for the declining efficiency advantage against the Joule cycle shown in Figure 8. This effect is even more pronounced for the TIT value of 1300 °C, for which the equivalence ratios are even lower.

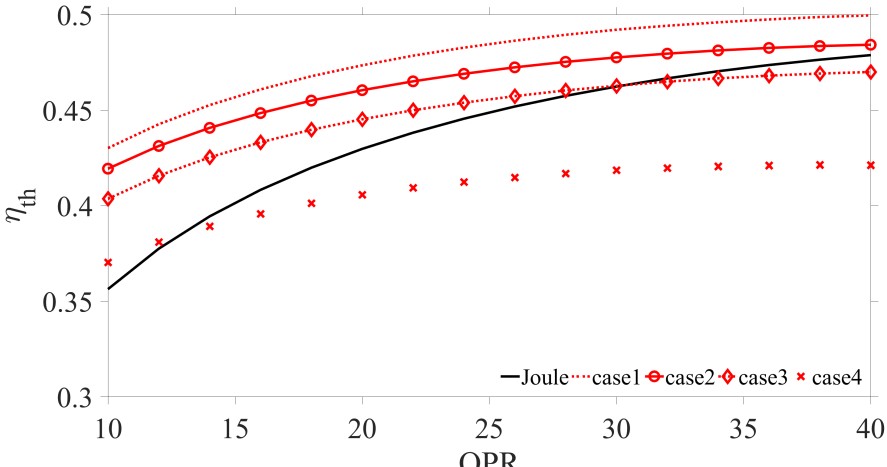

**Figure 8.** Cycle thermal efficiency as a function of compressor pressure ratio for the Joule cycle and the four versions of the Humphrey cycle presented in Table 3. TIT: 1700 °C.

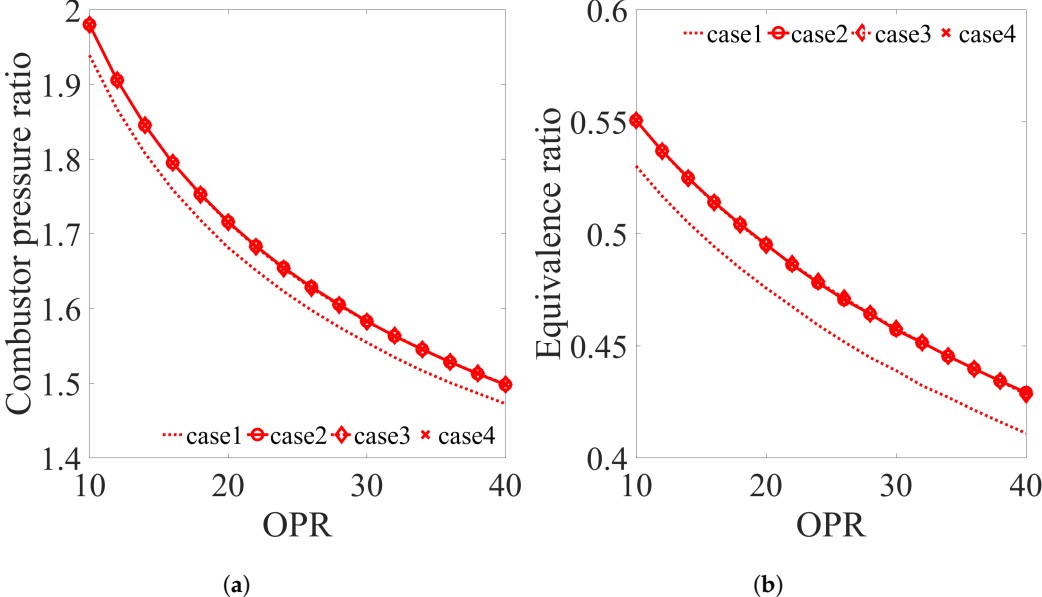

(**a**)                                                    (**b**)

**Figure 9.** Combustor pressure ratio and equivalence ratio as a function of the cycle pressure ratio for the four versions of the Humphrey cycle presented in Table 3 and a turbine inlet temperature of 1700 °C. (**a**) Combustor pressure ratio at TIT 1700 °C; (**b**) Global equivalence ratio at TIT 1700 °C.

In Figure 9, it is also obvious that, when part of the combustible mixture is consumed under constant pressure conditions, the equivalence ratio of the combustor is increased (compare $PGC_1$ to the other cases). The reason for that is again the fixed combustor outlet temperature. The part of the

mixture that burns under constant pressure conditions causes a lower temperature increase across the combustor. To compensate for that effect, the equivalence ratio of the whole process must change to richer values. This results to a higher specific heat input ($\frac{Q}{m}$ in Equation (1)) and hence to a higher pressure ratio across the combustor.

The last parameter that has an effect on the cycle efficiency is the amount of turbine cooling air. Figure 10 presents this amount as a percentage of the total air mass flow rate delivered by the compressor. The first thing to observe is the increase in cooling air flow rates for all cases of the Humphrey cycle when compared to that of the equivalent Joule cycle. The reason for this increase is the pressure gain over the combustor. An additional cooling compressor is used to bring the cooling air of the first stator row to the combustor outlet pressure. This leads to an increase at the cooling air temperature for this blade row and consequently to an increase in its mass flow rate, for the same blade temperature.

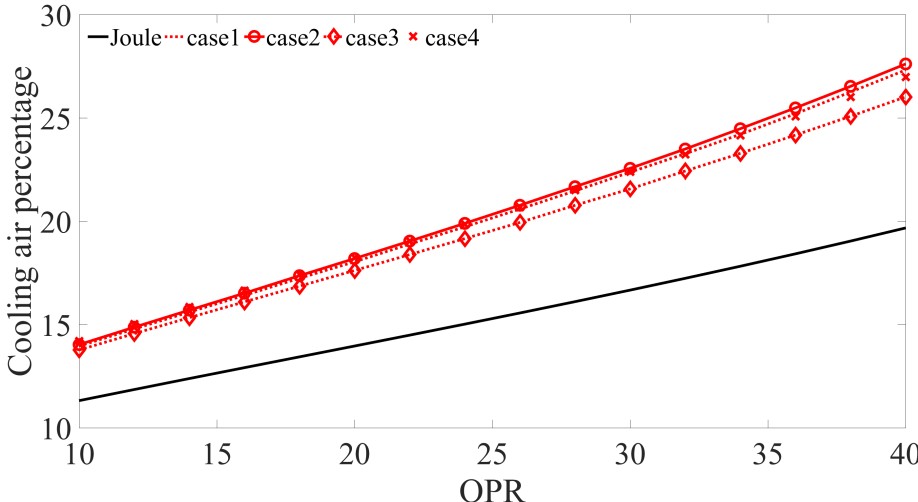

**Figure 10.** Percentage of compressor outlet air used for turbine cooling. TIT: 1700 °C.

A second observation is the higher increase rate of the cooling air mass flow for the Humphrey cycle for increasing cycle pressure ratios. This is an effect of the non linear dependency of the cooling air mass flow with its inlet temperature at the blades (see the models referred to in Section 2). This effect also considerably reduces the efficiency advantage of PGC cycle at higher compressor pressure ratios. A larger percentage of the turbine work must be consumed by the main cycle compressor to deliver cooling air to the turbine. This observation also underlines the necessity for more efficient cooling technologies if PGC is to be applied in gas turbines. Looking closer at the slight differences between the different simulation cases of the Humphrey cycle (see Table 3), one can make out the effect of all parameters changed between them. For example, we have already seen that the introduction of partly deflagrative combustion (from Case 1 to Case 2) results in a slight increase of the combustor outlet pressure. This in turn increases the outlet pressure and temperature of the cooling air compressor, which finally effects the slight increase of cooling air mass flow rate. Similarly, a larger pressure drop at the combustor inlet results in a lower combustor outlet pressure and thus lower cooling air temperature from this compressor. The outcome of this effect is clearly seen in Figure 10. Finally, the reduction of the first turbine stage efficiency (from Case 3 to Case 4) causes an increase of the outlet gas temperature from this stage and thus an increase of cooling air mass flow rate for the subsequent stage.

### 3.1.2. Specific Work Results

The second most important performance parameter for a cycle is its specific work generation. Especially for aero-engines, specific work is the first parameter used to characterize the size and weight of an engine that operates with the cycle in question. Up to date, pressure gain combustion has

been compared to constant pressure combustion on the basis of the specific impulse, mainly because pressure gain combustors are assumed to exhaust directly to atmosphere and not to a turbine expander. Figure 11 presents the results for the specific work generation of the studied systems, as work generated per mass of air compressed by the compressor.

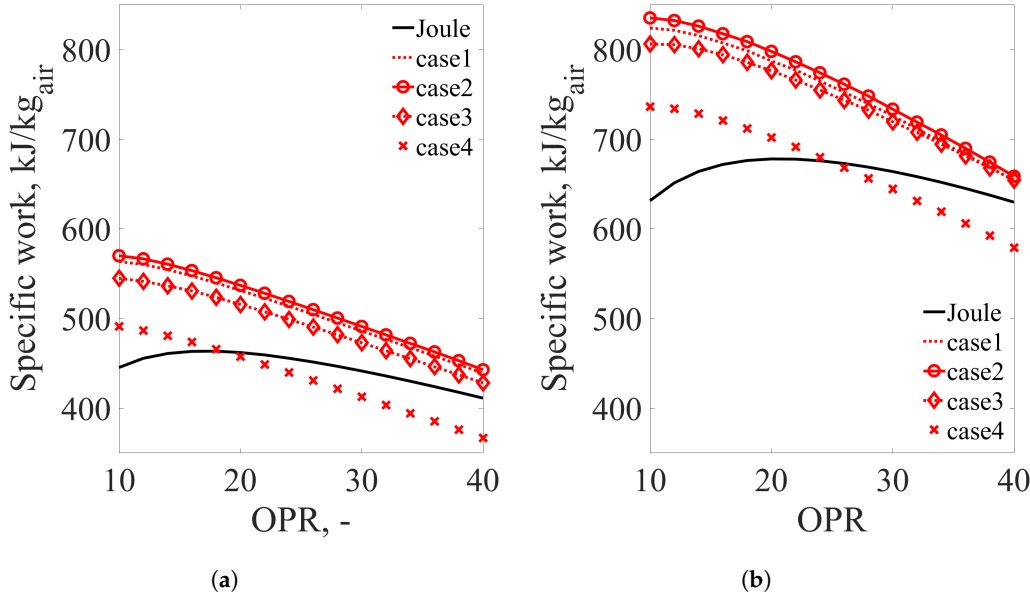

(**a**)　　　　　　　　　　　　　　　　(**b**)

**Figure 11.** Specific work as a function of pressure ratio for the Joule cycle and the four versions of the Humphrey cycle presented in Table 3 and two TIT values. (**a**) Specific work for TIT 1300 °C; (**b**) Specific work for TIT 1700 °C.

In Figure 11, it is clear that the Humphrey cycle has its maximum specific work at a slightly smaller pressure ratio than the lowest one captured from the current study. This is not the case for the Joule cycle, which demonstrates its maximum specific work value at a pressure ratio around 20 (slightly smaller than 20 for TIT = 1300 °C and slightly higher for TIT = 1700 °C). A second very important observation is that the Humphrey cycle has generally a higher specific work generation, if the first turbine stage isentropic efficiency is higher than 0.7. One interesting effect of partly deflagrative combustion (from Case 1 to Case 2 of Table 3) is the slight increase of the cycle specific work. This can be attributed to the slight increase of the equivalence ratio and the combustor outlet pressure caused by the introduction of deflagrative combustion in the cycle. Apart from this effect, the remaining observations are rather intuitive. An increase of the combustor inlet pressure drop or the turbine isentropic efficiency results in a reduction or an increase of the cycle specific work, respectively.

### 3.1.3. Gas Turbine Outlet Temperature

The outlet temperature of gas turbines is a very effective parameter to judge how well they can be coupled to a bottoming cycle, such as in the case of combined cycle power plants. Figure 12 shows the cycle outlet temperature for two TIT values and for all cases in Table 3. The first expected observation is that, for a given TIT value, the outlet turbine temperature is reduced for increasing pressure ratios. The same physical mechanism is responsible for the lower outlet temperature of almost all cases of the Humphrey cycle, compared to that of the equivalent Joule cycle. In fact, the pressure rise across the combustor results effectively in a higher turbine expansion ratio and thus to lower outlet temperatures for a given turbine inlet temperature. Only Case 4 for a TIT value of 1300 °C demonstrates a comparable turbine outlet temperature. However, this particular case also has the lowest thermal efficiency, as can be seen in Figure 7.

By looking at the temperature values, one can also mention that, despite their generally lower turbine outlet temperature, gas turbines operating on the Humphrey cycle could be combined with a

bottoming cycle for relatively low compressor pressure ratios. More specifically, one could look into the temperature values of the cycles with TIT 1700 °C (see Figure 12b). Heat recovery steam generators for combined cycles are typically fed with exhaust gas at temperatures of 550–700 °C. For the Joule cycle, this corresponds to a cycle pressure ratio of approximately 12–30. On the contrary, these temperatures are provided from the Humphrey cycle at pressure ratios of 12–30. Given the relatively high efficiency of the Humphrey cycle at these pressure ratios (see Figure 12b), one can conclude that a bottoming cycle might result to efficiency advantages. Such a study is beyond the aim of the current work and will be conducted in the future.

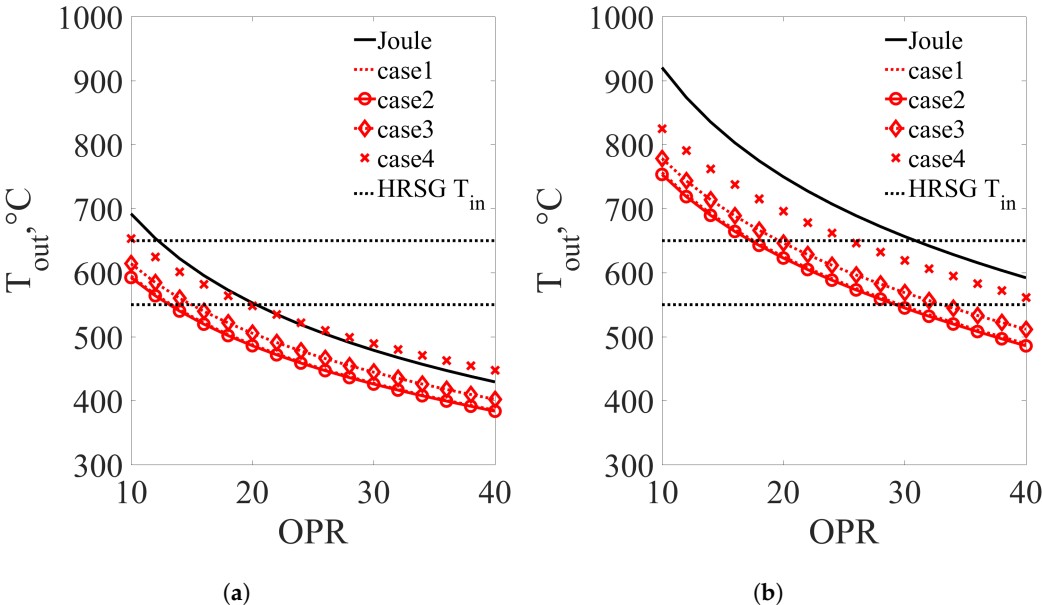

|     |     |
| :-: | :-: |
| (**a**) | (**b**) |

**Figure 12.** Turbine outlet temperature as a function of the cycle pressure ratio for the four versions of the Humphrey cycle presented in Table 3 and two TIT values. (**a**) Turbine outlet temperature for TIT 1300 °C; (**b**) Turbine outlet temperature for TIT 1700 °C.

### 3.1.4. On the Effect of the Cooling Air Compressor

There has been a lot of discussion in the PGC scientific community on the importance of the cooling air compressor and its power consumption for the thermal efficiency of PGC gas turbine cycles. In the current work, the cooling air compressor delivers air only to the first stator row. Figure 13a presents the work consumption of this compressor as a percentage of the work consumed by the main compressor of the cycle for a TIT value of 1700 °C. Generally, this value is approximately 1–3% of the work consumed by the main compressor. Looking more closely at Case 1, we see that this percentage follows a declining trend until a pressure ratio of 20, above which it stabilizes. Its value rises again after a pressure ratio of 30. This is a result of two counteracting phenomena. On the one hand, the increasing pressure ratio results in higher compressor outlet temperatures. This in turn increases the necessary amount of cooling air. On the other hand, the higher inlet temperature in the combustor leads to lower combustion equivalence ratios and thus to lower combustor outlet pressures (see Figure 9). This in turn results in a lower work consumption at the cooling air compressor. By comparing Case 1 with Case 2, one can conclude that the introduction of constant pressure combustion and the respective slight increase to the outlet pressure at the combustor outlet (see Figure 9) also leads to an increase in the work consumption of the compressor in question. Finally, from Case 3, it is understood that the introduction of pressure drop in the inlet of the combustor results, as expected, in a considerable decrease of the work consumption from the cooling air compressor.

Regarding the effect of the cooling air compressor on the cycle efficiency, Figure 13b presents the relative decrease in efficiency due to its work consumption. Here, a configuration of the cycle with and one without the compressor are compared. As expected, the efficiency of the cycle is decreased at most

for higher pressure ratios. Finally, one can see that the cooling air compressor has a stronger impact on the cycle efficiency, when the turbine efficiency is decreased (compare Cases 3 and 4 in Figure 13b). This is an additional reason much research must be carried out on the efficient harvesting of the energy in the exhaust gas of PGC combustor.

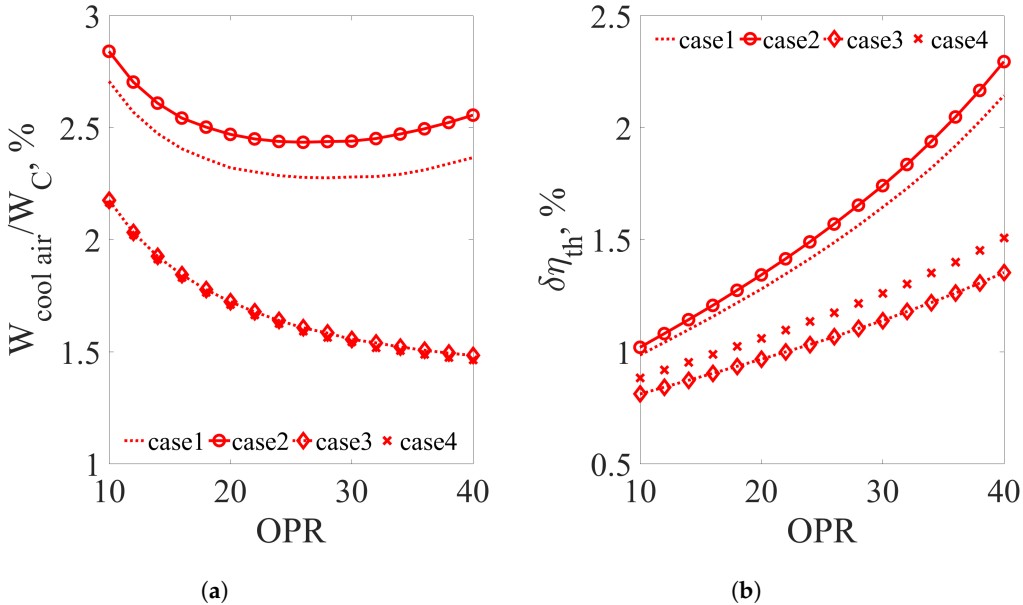

(**a**)    (**b**)

**Figure 13.** Cooling air compressor work and its effect on cycle efficiency as a function of the cycle pressure ratio for the four versions of the Humphrey cycle presented in Table 3. TIT: 1700 °C. (**a**) Cooling air compressor work as a percentage of main compressor work; (**b**) Cooling air compressor work effect on cycle efficiency.

### 3.1.5. On the Effect of Combustor Exhaust Gas Conditioning

As mentioned above, turbine integration is a crucial part of the research and development efforts towards the design of gas turbines with pressure gain combustion. In the preceding sections, several important cycle parameters and their effect on cycle performance have been analyzed under the assumption that the exhaust of a PGC combustor is directly fed to a turbine expander. Another approach to turbine integration is to install a combustor outlet/turbine inlet geometry that will condition the combustor exhaust flow and allow the operation of the turbine at its maximum isentropic efficiency. The current section explores this approach and aims to answer the question of allowable maximum losses for such a geometry.

This is done by additional simulations, where a combustor outlet geometry is modeled as a cooled turbine inlet guide vanes (IGV) row. In this case, the IGV row results in a pressure drop and the turbine stages downstream operate at their maximum isentropic efficiency (i.e., 0.9). The effect of this additional blade row on the cycle performance is studied with Case 3 (see Table 3) as its starting point. The pressure drop of the IGV row is subsequently increased until efficiency parity with an equivalent Joule cycle is reached. This is then defined as the maximum allowable loss of such a exhaust gas conditioning device.

By looking at the thermal efficiency of Case 3 (in Table 3) for the Humphrey cycle at a turbine inlet temperature of 1300 °C, it is concluded that it makes no sense to study the cycle at this condition. The Joule cycle will most probably be more efficient for all pressure ratios above 12. Because of this, the current analysis focuses on the Humphrey cycle with turbine inlet temperature of 1700 °C. Figure 14 presents the results for an increasing pressure drop of the IGV row, while every other cycle parameter is the same as for Case 3 in Table 3.

Figure 14 points to the fact that an IGV pressure drop of 20% has a comparable effect to a reduction of the first turbine stage isentropic efficiency by 20 percentage points. At the same time, it would be of

great advantage if conditioning devices could provide a stable exhaust flow for turbines with total pressure losses in the region of 5% of their inlet pressure. Even if the turbine inflow is not fully stable, a limited conditioning of the exhaust flow might still make a turbine design possible that will have high isentropic efficiency.

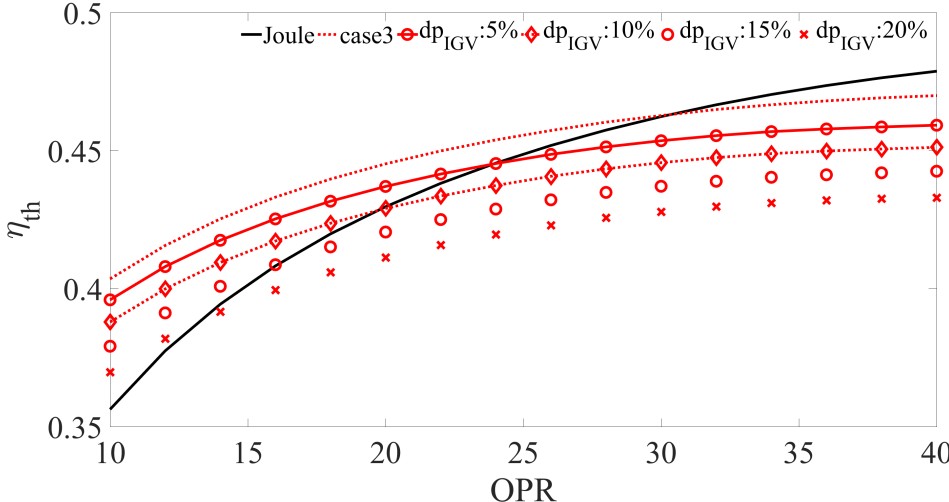

**Figure 14.** Cycle efficiency as a function of pressure ratio and IGV pressure drop for a TIT of 1700 °C.

### 3.2. Sensitivity Analysis

The preceding analysis is based on assumptions on the losses of several cycle components. For example, the inlet pressure loss and the mass percentage of the mixture combusted in a constant pressure manner are design features of the combustor. As a result, the analysis of the previous sections mainly enhances the understanding of the interdependencies of individual parameters and sets a benchmark for design goals of the components in question. In this context, it is very interesting to study the importance of each studied parameter for the resulting cycle efficiency and specific work. This was done through a sensitivity analysis. The sensitivity of the cycle efficiency and specific work on changes in the combustor inlet pressure drop ($dp_{CC}$), the turbine isentropic efficiency ($\eta_{isT}$) and the mixture mass percentage burned under constant pressure conditions ($m_{CPC}$) was studied. Each parameter was varied from a given reference value presented in Table 4. As one would expect, this sensitivity is also a function of the cycle pressure ratio (OPR) and turbine inlet temperature (TIT). To highlight and understand this effect, the aforementioned sensitivity analysis was performed for two values of the OPR and the TIT.

Figure 15 presents the sensitivity of the two cycle performance parameters for an OPR of 10 and a TIT of 1300 °C. The first thing to observe is that the turbine isentropic efficiency has the strongest impact both on the cycle efficiency and its specific work. This is an expected result, and underlines the importance of turbine performance for gas turbines with pressure gain combustion. In fact, a positive change of 20% in the turbine first stage isentropic efficiency results in a positive change of the cycle efficiency of 7.7%. Similarly, the cycle specific work is also increased by 7.8%. By comparing the impact of changes of the remaining two parameters, once can observe that $dp_{CC}$ has a stronger effect than $m_{CPC}$. More specifically, a 50% increase of $m_{CPC}$ results in 1.2% reduction of the cycle efficiency and 0.6% increase in its specific work. On the contrary, a 33% increase in $dp_{CC}$ causes a drop in efficiency of 1.4% and in specific work of 0.56%. As already mentioned in Section 3.1.1, the increase of specific work with increasing $m_{CPC}$ can be attributed to an increase in the combustor outlet pressure with increasing $m_{CPC}$. This is caused by the fact that the equivalence ratio must be increased in order to reach the same TIT with increasing $m_{CPC}$.

The results presented in Figure 15 reveal that, apart from designing turbines that can efficiently harvest energy from the exhaust of PGC combustors, it is very important to minimize the inlet pressure

drop in the latter. However, in many pressure gain combustors (mostly detonation based), this pressure drop is directly connected to the value of the third parameter of the presented sensitivity analysis, $m_{CPC}$. In fact, higher pressure drops generally tend to decrease $m_{CPC}$. Following this dependency, a designer must decide between specific work and efficiency and optimize the resulting cycle as a whole.

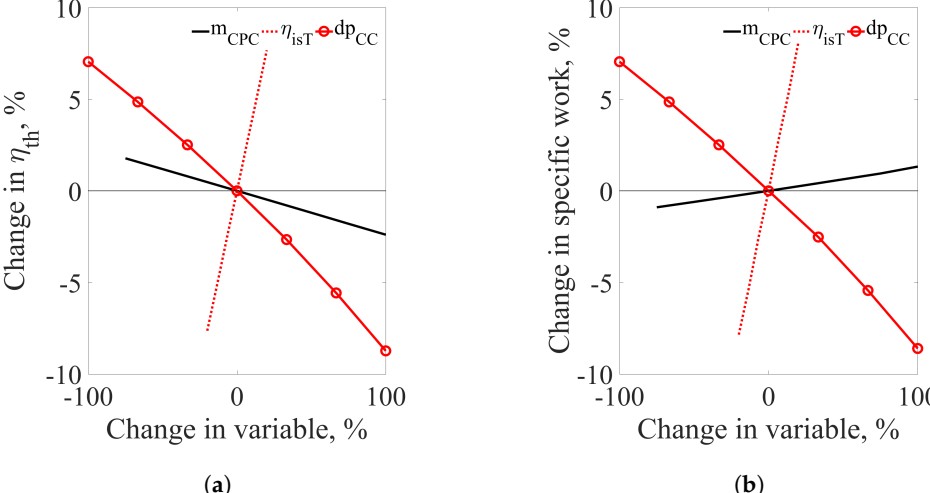

(**a**)                                                      (**b**)

**Figure 15.** Sensitivity of the cycle thermal efficiency and specific work on the three cycle parameters presented in Table 4. Pressure ratio: 10, TIT: 1300 °C. (**a**) Sensitivity of the cycle thermal efficiency; (**b**) Sensitivity of the cycle specific work.

Figures 16 and 17 present the effect of the turbine inlet temperature and the cycle pressure ratio, respectively, on the sensitivity analysis results. Both figures indicate that the cycle efficiency and specific work sensitivity on $m_{CPC}$ and $\eta_{isT}$ remain practically unchanged. The largest impact was observed on the effect of the combustor inlet pressure drop, which became less impactful for higher TITs and higher OPRs. This is expected, because a higher TIT also results in a high pressure gain across the combustor, due to the higher equivalence ratio in it. This in turn reduces slightly the effect of its inlet pressure drop.

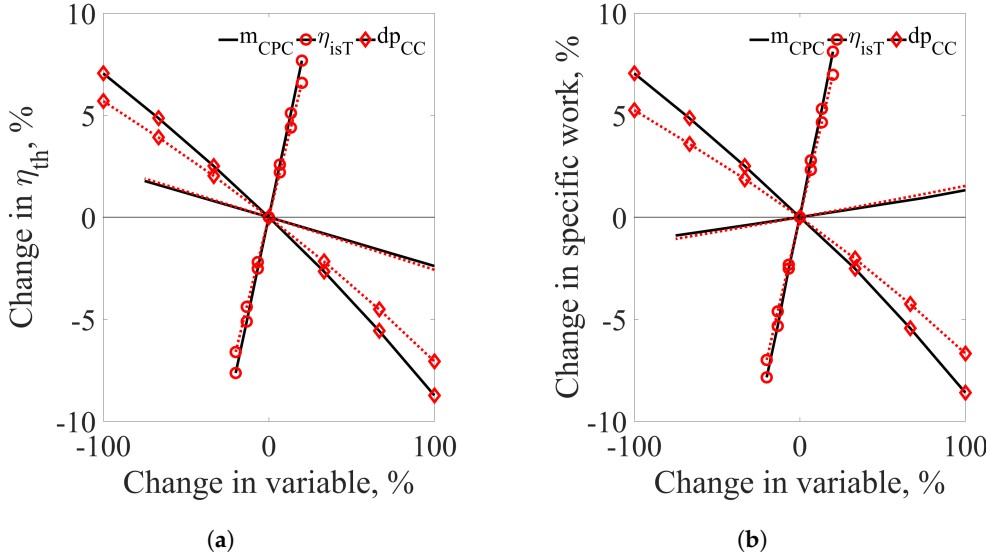

(**a**)                                                      (**b**)

**Figure 16.** Changes in the sensitivity of the cycle thermal efficiency and specific work on the three cycle parameters presented in Table 4, due to a different TIT value. Pressure ratio: 10. The continuous lines refer to a TIT = 1300 °C and the dotted lines to TIT = 1700 °C. (**a**) Sensitivity of the cycle thermal efficiency; (**b**) Sensitivity of the cycle specific work.

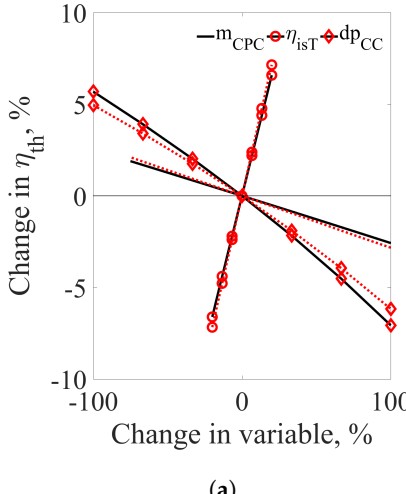 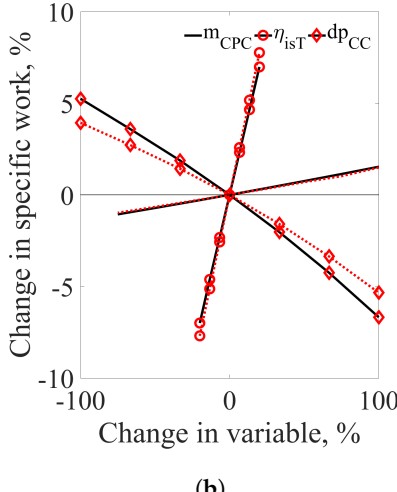

(**a**) (**b**)

**Figure 17.** Changes in the sensitivity of the cycle thermal efficiency and specific work on the three cycle parameters presented in Table 4, due to a different pressure ratio value. TIT = 1700 °C. The continuous lines refer to OPR = 10 and the dotted lines to OPR = 20. (**a**) Sensitivity of the cycle thermal efficiency; (**b**) Sensitivity of the cycle specific work.

## 4. Conclusions

This paper presents a comprehensive analysis of the Humphrey cycle for open cycle gas turbines with pressure gain combustion. To the author's knowledge, this is the first attempt to study all important cycle parameters and provide a benchmark for the performance of individual cycle components. The following list summarizes the most important takeaways of the current work.

- If realistic assumptions are made on the combustor inlet pressure drop, the turbine efficiency and the fuel mass consumed under constant pressure conditions, the Humphrey cycle will most probably make sense for mass averaged turbine inlet temperatures above 1500 °C and pressure ratios below 25.

- For a combustor inlet pressure drop of 15% and if 20% of the fuel mass is consumed under constant pressure conditions, the isentropic efficiency of the first turbine stage must be higher than 0.7. In any other case, the Humphrey cycle will not be able to compete with the Joule cycle of a similar technological level (i.e., turbine cooling effectiveness level).

- For the same cycle setting as before, the maximum pressure drop of an exhaust gas conditioning device that results in efficiency parity with the respective Joule cycle (provided that the turbine isentropic efficiency is 0.9) is approximately 15%. This parity is observed for this case at relatively low cycle pressure ratios. For higher pressure ratios, the allowable pressure loss is lower.

- For low cycle pressure ratios, the Humphrey cycle has a considerably higher specific work than the equivalent Joule cycle, even for low isentropic efficiencies of the turbine. Above pressure ratios of 20, this result changes if the turbine does not have an isentropic efficiency above 0.7.

- The Humphrey cycle generally results in lower turbine outlet temperatures for the same TIT with an equivalent Joule cycle. This is expected due to the higher turbine expansion ratio in the Humphrey cycle. This finding must be accounted for, when the Humphrey (but also other PGC gas turbine cycles) is considered the topping cycle of a combined cycle power plant.

- Generally, the Humphrey cycle (and to this effect most probably other PGC gas turbine cycles) has a considerably higher secondary air consumption than a Joule cycle of the same technological level. The fact that the cooling air of the first stator row has to be further compressed before its inlet in the turbine results in higher cooling air temperatures and thus increased secondary air consumption. This effect could be limited by compressor intercooling, cooling down the cooling air or improving the cooling effectiveness in the first stator row (i.e., improving the cooling technological level). It must be stressed here that the current study did not take into account the

possible increase of the convective heat transfer coefficient at the hot gas blade side. This might be the case due to unsteady flow phenomena that are not accounted for here. In this case, an even higher mass flow rate of cooling air would be necessary, for the same technological level.

- An initial study on the effect of the extra cooling air compressor showed that its integration results in an cycle efficiency decrease between one and three percentage points, depending on the cycle pressure ratio and settings.

Based on the presented analysis, all initial questions have been answered for the case of the Humphrey cycle with an ideal partial internal expansion in its constant volume combustion chamber. This cycle can effectively model pressure gain combustion processes such as pulsed resonant combustors and shockless explosion combustors. It also offers a rough approximation of the efficiency expected from detonation-based gas turbine cycles. For a more detailed study of the latter, another analytic combustor model, such as the one presented by Endo and Fujiwara [32], must be used. This model and its comparison to the current results will be the topic of a future study.

**Funding:** This research was funded by the Deutsche Forschungsgemeinschaft (DFG, German Research Foundation)-Projektnummer 200291049-SFB 1029.

**Acknowledgments:** The author gratefully acknowledges the support by the Deutsche Forschungsgemeinschaft (DFG) as part of the Collaborative Research Center SFB 1029 "Substantial efficiency increase in gas turbines through direct use of coupled unsteady combustion and flow dynamics".

**Conflicts of Interest:** The author declares no conflict of interest.

## Symbols

Latin Characters

| | | |
|---|---|---|
| $C$ | Turbine cooling technology level constant | |
| $c_v$ | Specific heat capacity under constant volume | $\frac{kJ}{kgK}$ |
| $c_p$ | Specific heat capacity under constant pressure | $\frac{kJ}{kgK}$ |
| $dp_{CC}$ | Combustor inlet pressure drop | % of $p_{in}$ |
| $dp_{IGV}$ | IGV pressure drop | % of $p_{in}$ |
| $K$ | Cooling air injection pressure loss constant | |
| $\dot{m}$ | Mixture mass flow rate | $\frac{kg}{s}$ |
| $m_{CPC}$ | Mixture mass percentage combusted under constant pressure | |
| $T_A$ | Combustor inlet temperature | K |
| $T_B$ | Combustor temperature at the end of the constant volume heat addition process | K |
| $T_3$ | Combustor outlet temperature | K |
| $p_A$ | Combustor inlet pressure | bar |
| $p_B$ | Combustor pressure at the end of the constant volume heat addition process | bar |
| $p_3$ | Combustor outlet pressure | bar |
| $Q$ | Heat added through combustion | W |
| $\delta p_{stage}$ | Expansion ratio of each turbine stage | |
| $\delta p_{mix}$ | Relative pressure drop due to cooling air mixing in the main exhaust stream | % of $p_{in}$ |
| $T_{bl}$ | Blade temperature used for cooling air calculations | K |
| $T_{out}$ | Turbine outlet temperature | K |
| $W_C$ | Main compressor work consumption | W |
| $W_{coolair}$ | Cooling air compressor work consumption | W |

Greek Letters

| | | |
|---|---|---|
| $\gamma$ | Specific heat capacity ratio | |
| $\varepsilon_F$ | Turbine stage film cooling effectiveness | |
| $\eta_{isC}$ | Compressor isentropic efficiency | |

| $\eta_{isC-cool}$ | Cooling air compressor isentropic efficiency | |
|---|---|---|
| $\eta_{Cooling}$ | Cooling air efficiency | |
| $\eta_{isT}$ | Turbine stage isentropic efficiency | |
| $\eta_{th}$ | Cycle thermal efficiency | |
| $\nu$ | Specific volume | $\frac{m^3}{kg}$ |
| $\pi_{CC}$ | Combustion chamber pressure loss coefficient | |
| $\pi_{losscool}$ | Turbine stage cooling pressure loss | |
| $\rho$ | Density | $\frac{kg}{m^3}$ |

## Abbreviations

| | |
|---|---|
| *CPC* | Constant pressure combustion |
| *CVC* | Constant volume combustion |
| *IGV* | Inlet guide vanes |
| *OPR* | Operational (compressor) pressure ratio |
| *PGC* | Pressure gain combustion |
| *PDC* | Pulsed detonation combustion |
| *RDC* | Rotating detonation combustion |
| *TIT* | Turbine Inlet Temperature |
| *ZND* | Zeldovich, von Neumann, Dörring |

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
