# Peer review of "Comprehensive Thermodynamic Analysis of the Humphrey Cycle for Gas Turbines with Pressure Gain Combustion"

_energies, doi:10.3390/en11123521_

Round 1

Reviewer 1 Report

File attached. 

Author Response

Dear reviewer, 

the changes I made to the text according to your comments are highlighted blue. 

Thanks for your effort and your comments

Reviewer 2 Report

The author presents in its paper a complete thermodynamic analysis of a gas turbine cycle using Pressure Gaining Combustion. A complete model of the gas turbine is constructed in aspen plus with a specifically constructed model for the PGC. By varying different parameters, the impact of several technical adaptations, necessary for the turbine to accept the hot gases coming from the PGC, and the impact of non-idealities could be assessed. The results identify clearly in what pressure and TIT range the PGC is favourable over the classical joule cycle. Finally, a sensitivity analysis has been conducted to study the impact of different

In general, the quality of the work is very high as well as the paper itself. The paper is nicely structured, including a complete introduction to the subject, clear expression of the novelty and the aim of the work. In addition, the model is clearly introduced, and all results are discussed in great details and support the conclusions. In my opinion, the paper can be accepted for publication if the authors adjust the minor comments listed below.   

Minor remarks:

The abstract and introduction are very clearly explained, especially the presentation of the different aims of the paper. However, on line 105, the author mentions that Aspen was used to model the gas turbine, while a few lines before, he discussed the issue of the non-continuous gas stream going into the turbine, making that actually a time dependent model should be used, while Aspen can only simulate steady-state. Although the author clearly explains later in the paper how this issue was handled (lines 238-250), I believe it would be better if this would already be addressed briefly here (or add a reference to later), since this raises some questions at this point.

The sentence line 133 is not completely clear and should be revised.

 Lines 146-148: without any proper introduction, it is not clear why for the part of the combustion that occurs at constant pressure, p3 is used, which is higher than the outlet pressure of the compressor. The authors should indicate why, even for the PGC-CPC, the pressure still increase compared to the compressor outlet.

Page 6, section 2.3 gas turbine model: It is not clear from the section whether the 3-stage turbine was modelled as one single block in aspen, with one isentropic efficiency or that it was modelled as a series of turbine blocks. This should be made clear. Additionally, what was the performance of the cooling air compressor is (efficiency and pressure ratio). Finally, from table 2, the reported efficiency of the turbine, is this the global efficiency or for each stage.

 Section 2.4 simulation procedure: Why was the maximal pressure ratio set at 40 (which is already very high)? Additionally, it is not clear which values where taken when simulating the Joule cycle (same values as reported in Table 2?)

 Section 3: I believe it would be better if the author refers to the different case in figures, that he would use the same names as used in the figure legends. This makes it easier for the reader to understand, without having to return each time to table 3.

 Lines 277-279: the sentence is written in a confusing way, especially by the specific start of the sentence ‘one cannot fail to observe’. I would propose to adjust this sentence, making it more clear.

Figure 6 and 7: it would be interesting if the data on these two figures could be merged in one figure (if possible to make it still clear), allowing to compare also the results at different TIT.

Lines 298-299: the fact that the PGC shows really good performance at low pressure ratios would indicate that using pressure ratios even lower than 10 would result to even bigger differences in terms of performance. This would mean that integrating PGC in smaller machines (even micro gas turbines) would be beneficial. I would like the author to comment on this.

Figure 10 is a bit misleading, since the scale on the y-axis is different. This can possibly be solved by putting them on the same figure, also allowing to compare specific work at different TIT (so impact of TIT).

 Lines 349-352: It seems from figure 10 that the maximal performance is around 7-8 for the PGC (depending on the case), so it would be interesting (for future work) to expand the simulations to even lower pressure ratios.

Author Response

Dear reviewer, 

thanks for your effort and your comments. I highlighted red the changes in the text that correspond to your comments

Best regards

Reviewer 3 Report

The author presents several results about the study of the Humphrey thermodynamic cycle in comparison with the classical Joule cycle for gas turbine performance analysis. The author includes in the model the expected impact of real-machine effects such as partial CPC, pressure losses, and turbine efficiency variation. The author tries (successfully) to answer to the main question about pressure-gain combustion applied to turbomachinery, that is if the great amount of efforts necessary to move towards that innovative solution is worth it.

The introduction to the problem is sufficiently accurate. The description of the methodology is clear and concise and the limitations of the selected approach are underlined. Results section is well organized. Conclusions are supported by the obtained results.

In conclusion, paper level is quite high and a few recommended changes follow:

- page 3 lines 88-93 the text does not fit the discussion and should be moved elsewhere. Maybe it could start with "The final aspect..." instead of "Finally..."

- page 3 line 96 "effect" should be replaced by "affects"

- page 7 lines 207-209 the meaning of the text is unclear

- page 8 Table 4 why did the reference value for turbine first stage is 75% instead of 70%?

- page 11 Figure 8 why does two images are prepared instead of one with Equivalence Ration VS Combustor PR?

- page 13 lines 366-367 the author should explain better the reason why increasing the OPR the T_out decreases

- page 13 lines 375-376 instead of 25-30 and 12-20 I would say 25-40 and 12-30. Including dashed lined in Figure 11b would help

- page 14 Figure 12a according to the author OPR can be divided in three: from 10 to 20 where the reduction of the specific work overcome the effect of an increased coolant mass-flow, a region between 20 and 30 where the contrary happens, an a region between 10 and 20 where the effects are balanced. This is not clearly explained in the text.

page 14 Figure 12b the definition of delta_eta_th,% is not clear and is is difficult to understand the concepts expressed in lines 398-404

- page 15 lines 447-448 why did the author used OPR 10 and TIT 1300°C? Selecting 1700°C would be more interesting, and OPR 10 is the lowest value.

Author Response

Dear reviewer, 

thanks for your effort and your comments. I highlighted the changes in thetext in green. 

Best regards
